# Platooning of Autonomous Public Transport Vehicles: The Influence of Ride Comfort on Travel Delay

**Teron Nguyen [1,2,3,]*, Meng Xie [2], Xiaodong Liu [2], Nimal Arunachalam [2], Andreas Rau [2], Bernhard Lechner [1], Fritz Busch [4] and Y. D. Wong [3]**

1   Institute of Road, Railway and Airfield Construction, Technical University of Munich, Baumbachstr. 7, 81245 Munich, Germany; bernhard.lechner@tum.de
2   Rapid Road Transport, TUMCREATE Ltd., 1 Create Way, #10-02 CREATE Tower, Singapore 138602, Singapore; meng.xie@tum-create.edu.sg (M.X.); xiaodong.liu@tum-create.edu.sg (X.L.); nimal.arunachalam@tum-create.edu.sg (N.A.); andreas.rau@tum-create.edu.sg (A.R.)
3   Centre for Infrastructure Systems, Nanyang Technological University, N1-01b-51, 50 Nanyang Avenue, Singapore 639798, Singapore; CYDWONG@ntu.edu.sg
4   Chair of Traffic Engineering and Control, Technical University of Munich, Arcisstr. 21, 80333 Munich, Germany; fritz.busch@tum.de
*   Correspondence: teron.nguyen@tum-create.edu.sg or teron.nguyen@tum.de; Tel.: +65-8376-1636

**Abstract:** The development of advanced technologies has led to the emergence of autonomous vehicles. Herein, autonomous public transport (APT) systems equipped with prioritization measures are being designed to operate at ever faster speeds compared to conventional buses. Innovative APT systems are configured to accommodate prevailing passenger demand for peak as well as non-peak periods, by electronic coupling and decoupling of platooned units along travel corridors, such as the dynamic autonomous road transit (DART) system being researched in Singapore. However, there is always the trade-off between high vehicle speed versus passenger ride comfort, especially lateral ride comfort. This study analyses a new APT system within the urban context and evaluates its performance using microscopic traffic simulation. The platooning protocol of autonomous vehicles was first developed for simulating the coupling/decoupling process. Platooning performance was then simulated on VISSIM platform for various scenarios to compare the performance of DART platooning under several ride comfort levels: three bus comfort and two railway criteria. The study revealed that it is feasible to operate the DART system following the bus standing comfort criterion ($a_y = 1.5$ m/s$^2$) without any significant impact on system travel time. For the DART system operating to maintain a ride comfort of the high-speed train (HST) and light rail transit (LRT), the delay can constitute up to $\approx 10\%$ and $\approx 5\%$ of travel time, respectively. This investigation is crucial for the system delay management towards precisely designed service frequency and improved passenger ride comfort.

**Keywords:** autonomous public transport; passenger ride comfort; travel time; horizontal alignment; microscopic traffic simulation

## 1. Introduction

The emergence of autonomous vehicles (AVs) has engendered innovative solutions for traffic congestion mitigation as well as the improvement of the passenger riding experience. The traveling public can expect level 5 full automation in more than 50% of vehicles by 2030 [1]. Herein, AVs can be readily operated as platoons on the streets with minimum gaps between individual AVs, thereby resulting in a significant increase of road capacity and improving fuel economy [2]. On the other

hand, by eliminating the driving tasks, vehicle occupants (drivers and passengers) can utilize on-board traveling time for activities such as reading, chatting or even working [3,4], which is expected to increase the productivity and enable other activities to be executed within a day [5]. For example, commuter services in motion are designed for NEXT's modular self-driving vehicles with built prototypes of autonomous pods in Dubai [6]. To achieve efficient mobility services, AV platooning in which consecutive vehicles conjugate as a road-train on the street is a good solution.

As for the on-road autonomous public transport (APT) system, which is a public transport mode that can guide itself without human conduction, there is a trend of connecting singular modules to form platoons on the road. This is the latest advance after the well-developed and implemented car platooning [7] and truck platooning [8] where a number of vehicles are traveling together and electronically connected. For example, recent research at TUMCREATE in Singapore is aimed at developing a dynamic autonomous road transit (DART) system at a much higher journey speed of autonomous bus (AB) platoons (at an average speed of 28km/h) than conventional buses (at an average speed of 19km/h) to offer a higher capacity level [9]. With a vehicle module of 6m length, 3.1 m height, 2.7 m width, and capacity of 30 passengers/module, the DART system is designed to flexibly adapt to passenger demand by electronically-linked platoons of the vehicles/modules on shared route segments and to decouple for route divergence. Relevant studies have been conducted focusing on scheduled platoon planning [10], fleet size estimation [11], and the deployment framework [12]. Similar high-speed platooning public transport can be found in Dubai under testing [6,13,14] as well as autonomous rail rapid transit in China (see Figure 1).

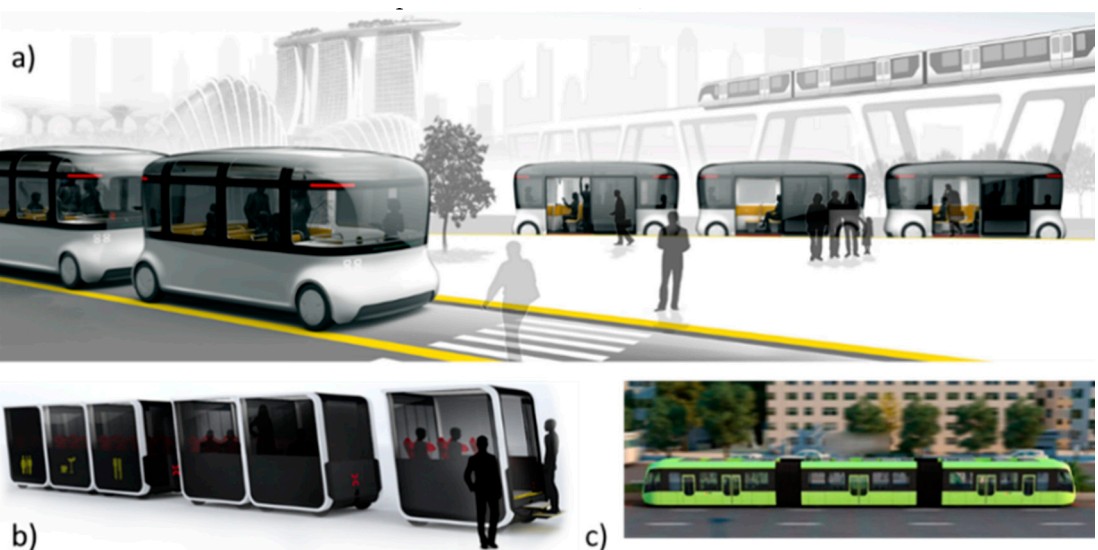

**Figure 1.** Examples of autonomous public transport (APT) platooning in (**a**) Singapore, source: https://www.tum-create.edu.sg/; (**b**) NEXT's modular self-driving vehicles designed in Dubai, source: http://www.next-future-mobility.com/; and (**c**) Autonomous rail rapid transit in China, source: http://www.crrcgc.cc/zzs.

Although automation may bring down the driver-cost in dense networks such as the urban context, the requirements of the schedule, fleet size, and route optimization are also raised [15]. The application of APT platooning in a large-scale operation has required a new concept in order to maintain a fixed timetabling frequency, e.g., every 5 minutes, for passenger transport. This is different from car platooning for private use or truck platooning for freight transport. Any deviation is expected to affect the system performance regarding travel time and speed, which reduces the whole APT system's reliability. Thus far, recent studies have focused on technological developments and often ignore the human factors which are of utmost importance in attracting car users to use public transport. The vehicle speeds are affected by various factors such as road geometrics, vehicle

performance and environmental conditions [16]. Considerably higher travel speeds are designed for the abovementioned APT systems (see Figure 1). However, there is always the trade-off between vehicle speed against rider comfort induced by the acceleration from the road surface/alignment and braking/accelerating. Hence, it is difficult to achieve ride comfort levels similar to high-speed trains (HST) and light rail transit (LRT), especially in a dense urban network with tight alignment and turning curves along the traveling routes. The comfort consideration is more critical for APT/AB as contrasted to (private) AV whereby AV passengers have greater discretion in their travel schedules and travel routes. They may be able to command the AV to run at the most comfortable speeds as well as along enjoyable routes (e.g., fewer turns or interruptions by intersections). On the other hand, APT/AB passengers often board and alight for shorter travel time and distance, and the APT/AB system must ensure its reliability (e.g., speeds, punctuality, and comfort). The question is raised as to which levels of platooning (e.g., average speed, number of coupling modules) can offer passengers the comfort levels of HST or LRT, or the lower comfort levels of conventional buses?

This study, therefore, aims to investigate the passenger-vehicle-road geometrics interaction to develop a new sustainable transportation system focusing on the user perspective. The platooning operation of the emerging DART system in Singapore city is considered as a case study. This study is part of a larger project at TUMCREATE to plan for a city-scale DART network [9]. The platooning protocol is first developed to simulate the coupling/decoupling process. Platooning performance is then simulated on PTV VISSIM platform for various scenarios to compare the performance of DART platooning under several ride comfort levels: three bus comfort levels [17] and two railway criteria. The horizontal alignment and passenger ride comfort are linked based on the back-calculation of vehicle speeds at different lateral acceleration levels. The study provides a new platooning protocol and comprehensive evaluation on the trade-off between passenger ride comfort against platoon performance e.g., travel time and platoon trajectories.

The remaining paper is structured as follows. In the literature review section, the relevant scientific literature is summarized. In the methods section, detailed steps in conducting the traffic simulation are described. The results and discussion section reports and discusses the main results of this investigation, as well as the outlook for further study.

## 2. Literature Review

Human factors are always a concern and consideration in the era of AVs [18]. Recent research has focused more on technological development such as platooning control [19], vehicle concept [20], cost efficiency [21], timetabling and scheduling [22], the experimental platform for vehicle control [23], mapping and path planning [24]. Apart from the AV, truck platooning has also attracted much research interest in [7,8,25], but there are very few studies on bus platooning. Regarding passenger perceptions, the empirical evidence from passenger security on the AB can be found in [26], or public attitudes towards AB in [27]. There is still minimal knowledge regarding the points-of-view of passenger ride comfort when developing these AV/AB systems.

Le Vine et al. [28] are perhaps the first researchers who dealt with the ride comfort of AV passengers by assuming that they can enjoy leisure activities as train passengers on a high-speed train (HST) [29] or light rail transit (LRT) [30]. Their assumption is premised on the fact that there is no existing empirical evidence on passenger perception aboard an operational AV. On the other hand, the ride comfort threshold used in [31,32] is more appropriate for car drivers, where lateral acceleration $a_y \leq 1.8 \text{m/s}^2$ is acceptable, $1.8 \text{m/s}^2 < a_y < 3.6 \text{m/s}^2$ is bearable, and $a_y > 5.0 \text{m/s}^2$ exceeds the human's bearing ability. It is noted that car passengers feel uncomfortable at lower acceleration levels compared to car drivers because passengers are not involved in active control of the steering wheel.

Indeed, there have been experimental studies on ride comfort and acceleration thresholds on the conventional bus. Regarding ride discomfort associated with longitudinal acceleration, researchers in [33] studied the discomfort thresholds due to the bus braking and speeding-up, in which bus passengers start to feel uncomfortable when longitudinal acceleration reaches $a_x > +1.5 \text{m/s}^2$ and the

deceleration/braking $a_x < -0.75 \text{m/s}^2$. Another study in France [34] analyzed the discomfort feeling of standing passengers regarding the bus interior design as well as the bus lane design. There are two levels of ride discomfort: Level 1 (uncomfortable) and Level 2 (loss of balance). Recently, a study in [17] surveyed the ride comfort of passengers at multiple postures aboard buses and suggested comfortable acceleration thresholds for the regular bus as well as for the future AB. The vehicle speeds at the discomfort threshold (uncomfortable at $a_y = 1.5 \text{ m/s}^2$) could be suggested when there is a high number of standing passengers while the vehicle speeds at a great discomfort threshold (very uncomfortable at $a_y = 1.75 \text{ m/s}^2$) could be suggested in case of all seated passengers. In any case, the vehicle speeds at the extreme discomfort threshold (extremely uncomfortable at $a_y = 2.0 \text{ m/s}^2$) must be avoided. The recent literature regarding typical ride comfort thresholds on various modes of transport are summarized in Table 1.

**Table 1.** Ride comfort thresholds at multiple directions.

| Source | Longitudinal Acceleration $a_x$ (m/s$^2$) | Lateral Acceleration $a_y$ (m/s$^2$) Acceleration Rate of Change C (m/s$^3$) | Transport Mode | Passenger Posture |
|---|---|---|---|---|
| [29] | $a_x = +1.34$: max acceleration $a_x = -1.34$: max braking | $a_y = 0.98$–1.47: uncomfortable | Light rail | Not specific |
| [30] | $a_x = +0.58$: max acceleration $a_x = -0.54$: max braking | $a_y = 0.49$: uncomfortable | Heavy rail | Not specific |
| [31,32] | | $a_y \leq +1.8$: acceptable, $a_y = +1.8 \div 3.6$: bearable $a_y > +5.0$: bearing ability | Car | Sitting |
| [35] | $a_x = -3.4$: comfortable braking | $a_y = 0.4$–1.3: safety within spiral curve $C = 0.3 \div 0.9$: comfortable rate of change | Car | Sitting |
| [28] | | $a_y = 1.47$: uncomfortable on light rail $a_y = 0.49$: uncomfortable on heavy rail | AV | Sitting |
| [36] | | $a_y = 0.6$–1.0: uncomfortable $C = 0.3$–0.6: uncomfortable | Guided bus | Not specific |
| [33] | $a_x > +1.5$: uncomfortable $a_x < -0.75$: uncomfortable | | Bus | Sitting |
| [34] | $a_x < -1.4, a_x > +1.5$: level 1 $a_x < -2.2, a_x > +2.5$: level 2 | $a_y < -1.4, a_y > +1.6$: level 1 $a_y < -2.0, a_y > +2.0$: level 2 | Bus | Standing |
| [17] | | $a_y \leq 1.5$: comfortable $a_y = 1.5 \div 1.75$: uncomfortable $a_y = 1.75 \div 2.0$: very uncomfortable $a_y > 2.0$: extremely uncomfortable | Bus, AB application | Sitting, leaning standing |

Note: Level 1, Level 2: uncomfortable, and loss of balance.

Table 1 shows that only one study [28] investigated passenger ride comfort on the AV versus the levels of service at an intersection, with the study's limitation of a small-scale intersection. Vehicle platooning as the main advantage of AV technology has not been considered, neither was any bus ride comfort criterion included. The attainment of ride comfort levels on a train, a transport mode that has dedicated railway running at higher speed levels, is challenging in an urban context, especially for APT/AB with features of frequent stop-and-go and turning at the intersection. Herein, this study overcomes these limitations by: (1) developing a new platooning protocol for APT coupling/decoupling; (2) simulating a long corridor with several intersections for APT platooning from 3 to 5 modules; and (3) investigating the trade-off between DART platooning performance against passenger ride comfort on the bus and train.

## 3. Materials and Methods

Researchers have used PTV VISSIM (PTV Group, Karlsruhe, Germany, https://www.ptvgroup.com/en/) as a reliable platform for microscopic traffic simulation and generating plausible results of incidents for evaluating system performance [37–39]. Herein, PTV VISSIM can generate vehicle trajectories for detailed analysis. With the considerable functionality of driving behavior modeling, PTV VISSIM with the external driver model (EDM), was chosen to develop many traffic control strategies for the AV [40] or cooperative adaptive cruise controls [41]. These capabilities have motivated this current study to use EDM for simulating DART platoons in a real road network.

### 3.1. Development of Coupling/Decoupling Protocol Based on EDM

There is minimal available information on the coupling/decoupling process for APT platooning following a timetable with a fixed frequency and fixed-route that can well cater to passenger demand. The APT platooning was developed and illustrated with its operational dynamics in Figure 2, where vehicles/modules from two different lines/branches (Step 1) couple/merge at a pre-defined stop (Step 2) and run together along their shared-routes/trunks (Step 3) before splitting/decoupling/diverging to their destinations (Step 4). The merged-platoons can also be formulated from shorter platoons, and the merged-platoons split once completing their shared-routes. This merging/splitting process is different from truck platooning problem in [42], in which the trucks are able to merge and split while running at a high speed. The fleet size model was studied by [11] while the deployment planning was investigated in [12], resulting in the timetable input for the system operation.

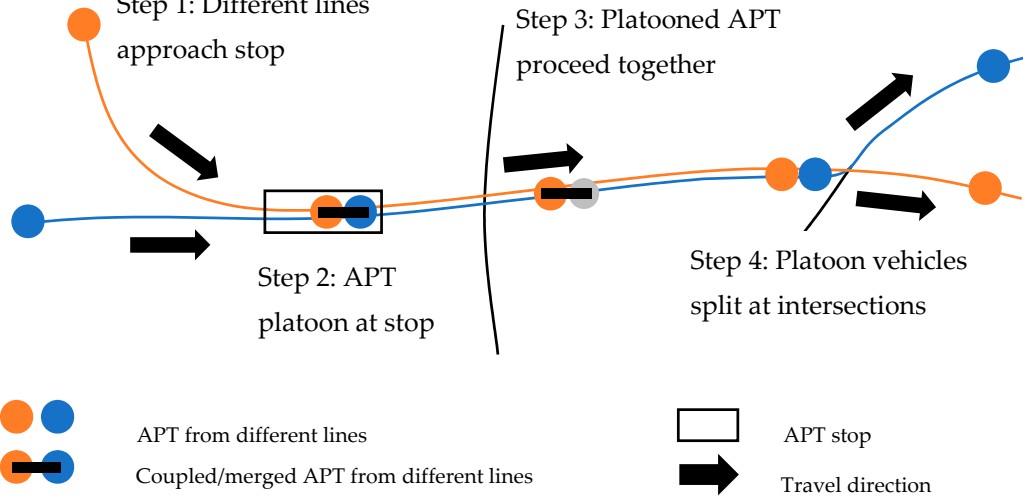

**Figure 2.** Illustration of the APT coupling based on bus platooning [43].

The coupling/decoupling protocol was developed with three main layers, namely strategic planning, tactical operation, and local behavior. Strategic planning follows the conventional public

transport planning but is extended with a coupling timetable which includes the time, location and line sequence of coupling. The departure times of all lines to be coupled together were adjusted to realize simultaneous arrivals at planned stops for coupling. Tactical operation supervises all APT vehicle operation in the network to guarantee the planed coupling, and to manage the cruising, dwelling of APT vehicles in case of both normal and delayed situation. Local behavior refers to autonomous driving behavior, and it complies to strategic planning and tactical operation. Local behavior corresponds to the consideration of passenger comfort and is developed based on the enhanced intelligent driver model (EIDM).

The local behavior was coded and interfaced with PTV VISSIM via the external driver model (EDM). There are five traffic flow conditions being considered by EIDM, namely free traffic, upstream jam front, congested traffic, downstream jam front and bottleneck sections [44]. The essential behavioral parameters in this study were the desired time gap T = 1.5 s, the desired maximum acceleration a = 1.3 m/s$^2$ and the desired deceleration b = 1.5 m/s$^2$. These acceleration and deceleration levels are based on the technical specification of the DART vehicle. Table 2 shows the $\lambda_T$, $\lambda_a$ and $\lambda_b$ as multiplication factors in different traffic flow conditions for the EIDM.

**Table 2.** Driving strategy matrix [45].

| Traffic Condition | $\lambda_T$ | $\lambda_a$ | $\lambda_b$ | Driving Behavior |
|---|---|---|---|---|
| free traffic | 1 | 1 | 1 | default/comfort |
| upstream front | 1 | 1 | 0.7 | increased safety |
| congested traffic | 1 | 1 | 1 | default/comfort |
| downstream front | 0.5 | 2 | 1 | high dynamic capacity |
| bottleneck | 0.7 | 1.5 | 1 | breakdown prevention |

### 3.2. The Effects of Ride Comfort Criteria on DART Performance

After the platooning protocol was established, different scenarios were considered to evaluate the DART performance as follows (see the summary in Table 3). The long corridor included several intersections where the merged-platoons must navigate along sharp turning curves (see Figure 3). Apart from LRT and HSR ride comfort criteria, the other three lateral thresholds regarding passenger posture onboard [17] were also considered. For longitudinal acceleration and comfort, bus deceleration/braking $a_x = -0.75 \text{m/s}^2$ [33] was used to define the reduced speed areas, which is much lower than the desired deceleration *b* of the designed vehicle. Each merged-platoon included 3 to 5 modules running from start to end, where the starting point was a pre-defined merging stop, and the ending point was the last stop before decoupling. In this study, vehicle dynamical behavior within curves was the focus by using microscopic traffic simulation, where three scenarios were created with the merged-platoons consisting of 5, 4 and 3 modules. All scenarios were developed without traffic interference which can be considered as an ideal public transport prioritization scenario with no delay caused by the traffic light. The operating speed was 49 km/h on straight segments.

**Table 3.** Simulation scenarios.

| Merged Platoon | Number of Modules | Platoon Formation From | | Ride Comfort Criteria and Lateral Acceleration Thresholds | | | | | Traffic Conditions |
|---|---|---|---|---|---|---|---|---|---|
| | | Platoon A | Platoon B | HST Comfort $a_y = 0.49$ m/s$^2$ | LRT Comfort $a_y = 0.98$ m/s$^2$ | Bus Standing $a_y = 1.50$ m/s$^2$ | Bus Leaning $a_y = 1.75$ m/s$^2$ | Bus Sitting $a_y = 2.0$ m/s$^2$ | Dedicated Lane without Traffic Interference |
| Platoon 1 | 5 | 2 | 3 | ✔ | ✔ | ✔ | ✔ | ✔ | ✔ |
| Platoon 2 | 4 | 2 | 2 | ✔ | ✔ | ✔ | ✔ | ✔ | ✔ |
| Platoon 3 | 3 | 2 | 1 | ✔ | ✔ | ✔ | ✔ | ✔ | ✔ |

A small corridor including 3 intersections and 5 turning curves with different radii (Figure 3b,c) was extracted from the more extensive network that consisted of 18 DART lines, 5965 street sections with a total length of over 670km. In the planned DART network (Figure 3a), there are numerous turning curves which are different from the highway whose horizontal alignment is designed with larger curve radii that facilitate the formation of AV platoons. With different levels of lateral acceleration ($a_y$), vehicle speeds ($V$ in km/h) can be back-calculated based on turning movements and curve radius ($R$ in m) as Equation (1):

$$V = 3.6\sqrt{a_y R} \tag{1}$$

This estimation is inferred from the basic equation that governs vehicle operation on a curve following the physical laws of motion [35]. The effects of a lateral jerk and turning duration have been neglected in this simple calculation. The proposed speeds can be used as the speed limit for AB along the corridors (see Table 4) as an important input for the reduced speed areas in VISSIM. The logic is that for new innovative APT systems coupled with the developed navigation technology, APT will be able to detect road geometry, curves and turning movement trajectory at any location along its route. Instead of using ArcGIS to measure distance [46], this study analyzed turning radii in AutoCAD after extracting coordinates of the travel corridor from Google Maps. Indeed, road horizontal alignment can be identified using mobile mapping systems and GIS spatial data as the investigation in [47,48].

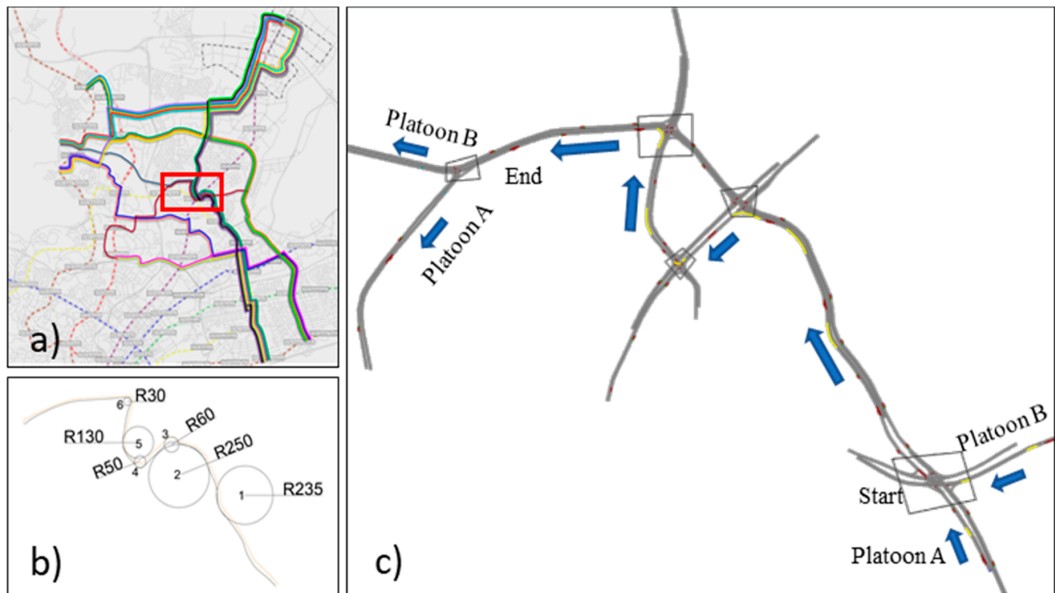

**Figure 3.** Planned DART network in Singapore (**a**) 18 lines; (**b**) extracted corridor consisting of 6 turning curves with stated radii as 6 reduced speed areas as input for traffic simulation; and (**c**) the formation of merged-platoons from platoon A and platoon B.

**Table 4.** Reduced speeds within curves, given the designed speed of 49km/h on straight segments.

| Curve Order | Radius (m) | HST Comfort $a_y$ = 0.49 m/s² | LRT Comfort $a_y$ = 0.98 m/s² | Bus Standing $a_y$ = 1.50 m/s² | Bus leaning $a_y$ = 1.75 m/s² | Bus Sitting $a_y$ = 2.0 m/s² |
|---|---|---|---|---|---|---|
| 1 | 235 | 34 | 48 | 59 (49) | 64 (49) | 68 (49) |
| 2 | 250 | 35 | 49 | 61 (49) | 66 (49) | 70 (49) |
| 3 | 60 | 17 | 24 | 30 | 32 | 34 |
| 4 | 50 | 16 | 22 | 27 | 29 | 31 |
| 5 | 130 | 25 | 35 | 44 | 47 | 51(49) |
| 6 | 30 | 12 | 17 | 21 | 23 | 24 |

Note: value inside the "()" is used once the calculated speed exceeds the designed speed of 49 km/h.

## 4. Results and Discussion

### 4.1. Platooning Behaviors and Trajectories

For each simulation scenario among the five ride comfort levels, the three platoon 1, platoon 2, and platoon 3 started at different time steps of 180s, 480s, and 780s, respectively, following a frequency of 5 minutes (see Table 5). In order to meet at the first stop to form platooning along the shared-route/trunk, the two platoons A and B departed much earlier from two branches/lines. Two additional terminals were allocated for individual modules to form platoons A and B (see Figure 4).

The trajectories of each module in platoons at different lateral acceleration levels along the corridor are shown in Figure 4, where the first stop is at the distance = 0 (m). The R software package was utilized for data processing. Two platoons A and B had merged at the first stop and traveled together to the end before splitting into two different destinations. The trajectories are quite similar even with the composition of 3, 4, or 5 modules. This has demonstrated the efficiency of the developed platoon protocol, in which the follower(s) always try to catch up with the leader according to specific conditions of the desired time gap (T = 1.5 s), the desired maximum acceleration (a = 1.3 m/s$^2$) and the desired deceleration (b = 1.5 m/s$^2$). Although different from car or truck platooning, APT platoons/modules must frequently dwell at stops for boarding and alighting passengers, as well as at signalized intersections whenever traffic light is not in its favor.

**Table 5.** Starting time and arrival time of three merged-platoons (unit: time step in second).

| Merged Platoon | Start (s) | Arrival (s) | | | | |
|:---:|:---:|:---:|:---:|:---:|:---:|:---:|
| | | **HST Comfort** | **LRT Comfort** | **Bus Standing** | **Bus Leaning** | **Bus Sitting** |
| 1 | 180 | 560 | 530 | 520 | 518 | 516 |
| 2 | 480 | 875 | 843 | 832 | 830 | 828 |
| 3 | 780 | 1170 | 1145 | 1130 | 1128 | 1126 |

Importantly, the effect of 5 levels of passenger comfort on DART travel time is shown based on the platoon trajectories. Due to the constraint of lateral acceleration, the designed speeds within curves are reduced substantially in cases of HST and LRT comfort criteria (see Table 4), especially along curve radii less than 100m (curves 3, 4 and 6). The delay gaps were cumulated by the travel distance after negotiating each curve and reached a maximum value at the ending stop. In Figure 4, a close-look at platoon 1 trajectories within curve 3 (R = 60 m) has shown an identical pattern of 5 modules within the platoon, but large differences between HST, LRT comfort criteria ($a_y$ = 0.49, 0.98 m/s$^2$) and bus comfort thresholds ($a_y$ = 1.5, 1.75 and 2 m/s$^2$).

Figure 5 presents the detailed information of 12 modules when they all appeared in the network. At the time-slice C-C, Platoon 1 (No = 1, 2, 3 from platoon B and 4, 5 from platoon A) was decoupled, while platoon 2 (No = 6, 7 from platoon B and 8, 9 from platoon A) were formed (in_platoon = 1) whereas platoon 3 (No = 10 from platoon B and 11, 12 from platoon A) has not been formulated yet (in_platoon = 0). This status is represented in the "speed" information, in which the identical velocity of ≈ 49 km/h is shown for platoon 1, and ≈ 6 km/h for platoon 2, whereas random speed levels are shown for platoon 3.

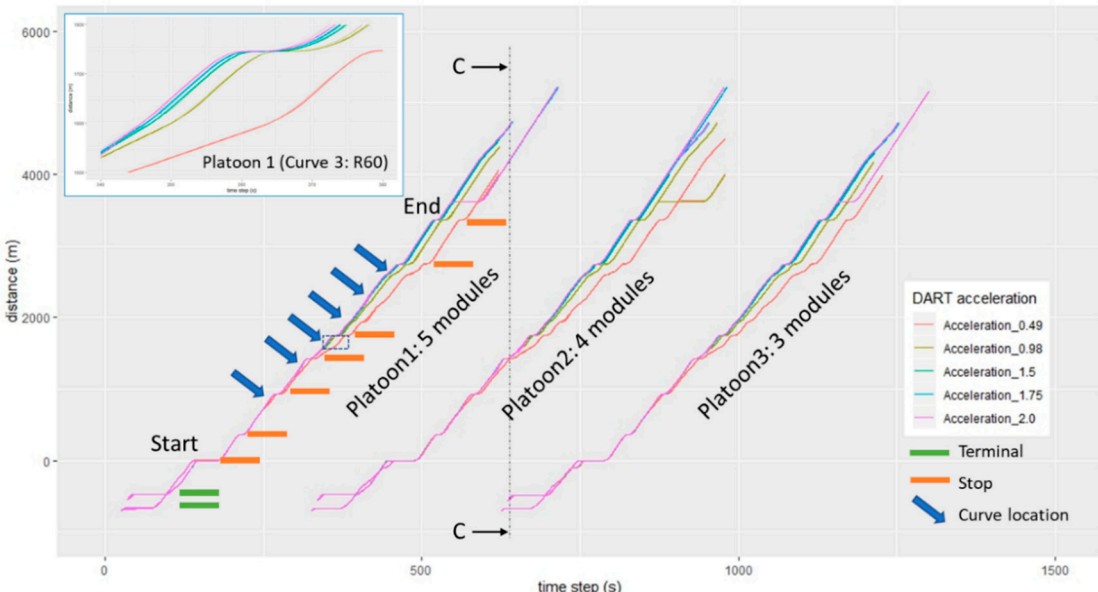

**Figure 4.** Trajectories of different merged-platoons of 3, 4, 5 modules. The locations of terminals, stops and curves along platoon 1 trajectories are also applied for platoon 2 and platoon 3.

| Count: 1 | No | VehType | Lane | Pos | Speed | DesSpeed | Acceleration | LnChg | PTLine | intacstate | dwell | platoonlen | pos_in_platoon | in_platoon | platoon_id |
|---|---|---|---|---|---|---|---|---|---|---|---|---|---|---|---|
| 1 | 1 | 700: srt | 104 - | 437. | 48.88 | 48.92 | 0.00 | None | 1 | 9.00 | | 3 | 1 | 1 | 1 |
| 2 | 2 | 700: srt | 104 - | 429. | 48.88 | 48.97 | 0.00 | None | 1 | 9.00 | | 3 | 1 | 1 | 1 |
| 3 | 3 | 700: srt | 104 - | 422. | 48.88 | 48.34 | 0.00 | None | 1 | 9.00 | | 3 | 1 | 1 | 1 |
| 4 | 4 | 700: srt | 15 - | 101 | 46.37 | 48.64 | 0.36 | None | 2 | 9.00 | | 2 | 2 | 1 | 2 |
| 5 | 5 | 700: srt | 15 - | 100 | 46.34 | 48.50 | 0.38 | None | 2 | 9.00 | | 2 | 2 | 1 | 2 |
| 6 | 6 | 700: srt | 7 - 4 | 114. | 6.19 | 48.70 | 1.15 | None | 1 | 9.00 | | 2 | 1 | 1 | 3 |
| 7 | 7 | 700: srt | 7 - 4 | 107. | 5.40 | 48.16 | 1.00 | None | 1 | 9.00 | | 2 | 1 | 1 | 3 |
| 8 | 8 | 700: srt | 7 - 4 | 100. | 5.40 | 48.96 | 1.00 | None | 2 | 9.00 | | 2 | 2 | 1 | 4 |
| 9 | 9 | 700: srt | 7 - 4 | 94.2 | 5.40 | 48.95 | 1.00 | None | 2 | 9.00 | | 2 | 2 | 1 | 4 |
| 10 | 10 | 700: srt | 87 - | 48.9 | 0.12 | 48.59 | -0.03 | None | 1 | 9.00 | | 1 | 1 | 0 | 5 |
| 11 | 11 | 700: srt | 35 - | 11.9 | 47.04 | 48.18 | -0.68 | None | 2 | 9.00 | | 2 | 2 | 0 | 6 |
| 12 | 12 | 700: srt | 35 - | 8.64 | 29.82 | 48.75 | -6.37 | None | 2 | 9.00 | | 2 | 2 | 0 | 6 |

Vehicle Types | Public Transport Line... | Scripts | Vehicle Classes / Veh... | **Vehicles In Network** | Public Transport Stop... | Maximum Decelerati... | Desired Deceleratio

**Figure 5.** Screenshot from VISSIM shows detailed information of vehicles all appeared at the time-slice C-C in Figure 4. The platooning information is illustrated based on the under-developed coupling/decoupling protocol.

### 4.2. Travel Time and Delay of DART Platoons

To evaluate the system performance of this APT system, the travel time and delay of the three platoons are summarized in Figure 6. The travel time for each merged-platoon (platoon 1, platoon 2, and platoon 3) is calculated when all modules in the platoon depart from the pre-defined stop (start point) to the final stop (endpoint). For the travel distance of 3.6 km, it takes roughly 340s ÷ 390s (equivalent to travel speeds of 33.23 ÷ 38.11 km/h) for the merged-platoons to finish the shared-route. As for the delay results, the travel time from the ride comfort criterion of a sitting bus is considered as a reference for comparison (see Equation (2)). The simulation scenario without traffic interaction can be considered as an ideal condition of traffic signal prioritization, which enables the platoons to run freely from start to end. The delay ratio is calculated as Equation (3):

$$Delay_i = Travel\ time_i - Travel\ time_{bus\ sitting} \tag{2}$$

$$Delay\ ratio_i = \frac{Delay_i}{Travel\ time_i} \tag{3}$$

The platooning protocol has formulated merged-platoons running as designed speeds along the corridor. Overall, the travel time and delay of merged-platoons following the ride comfort levels on the bus (bus standing, bus leaning, and bus sitting) are quite identical, meaning that it is feasible to operate the platoons following the bus standing comfort with lateral acceleration $a_y = 1.5$ m/s$^2$. On the other hand, the proportion of the delay is significant at $\approx$10% and $\approx$5% of travel time of all platoons following HST and LRT comfort criteria, respectively. It is noted that the platooning protocol is designed with maximum waiting time, e.g., of 60s, and the travel delay as 44s for a 3.6 km corridor (as shown in Figure 6) can be extended for a longer travel distance, which can deteriorate the pre-defined platooning sequence. This issue would be scaled up to the larger planned network of 18 lines, resulting in delay effects for the whole system.

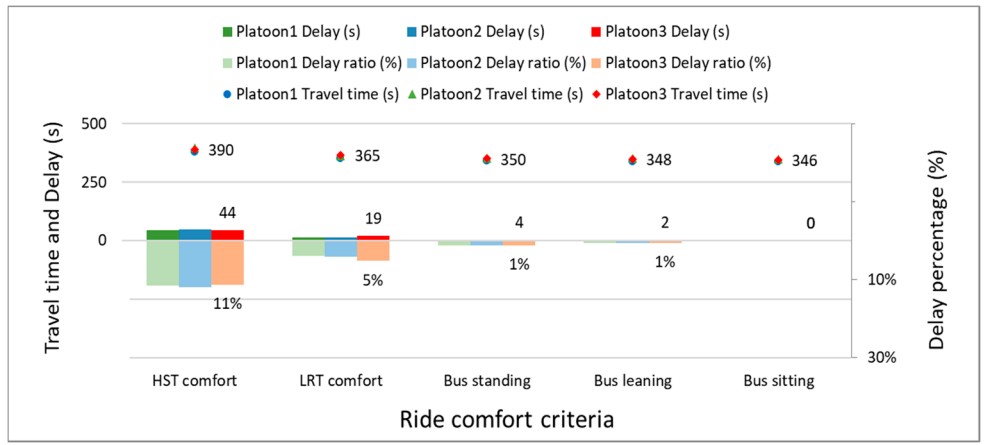

**Figure 6.** Travel time and delay (bus sitting as reference) of 3 platoons at different ride comfort criteria. The value inside the graphs is the travel time and delay of platoon 3 for reference.

Under different scenarios, the results have shown that the performance of DART platooning is influenced by turning curves, mainly the sharp turning curve with a radius less than 100m with significantly reduced speed constraints. The module number was varied to evaluate the dynamic effects of short and long platoons on the travel time. The results have shown that the number of modules, either 3, 4 or 5 in the merged-platoons, does not affect system performance (e.g., travel time, delay). However, the influence of ride comfort on travel delay is critical in an urban context when operating the DART system following HST and LRT comfort criteria. This investigation in passenger-vehicle-road geometrics interaction is crucial for the delay management of the system, towards the precisely designed timetable/frequency within the trade-off between system performance and passenger ride comfort.

With technological advancements, the emergence of autonomous public transport (APT) is ongoing worldwide towards the finalization of level 5 automation. Autonomous cars can provide many benefits and attract more drivers and passengers. However, this may also promote a car-dependent society. Therefore, how to improve the APT service quality to attract more passengers for using this innovative transport mode is of utmost importance that motivated this study. At first, the platooning protocol was developed in this study to support the new concept of electronically coupling/decoupling of APT platooning from 3 to 5 modules at multiple departures with a frequency of 5 minutes. The VISSIM EDM was utilized as an essential platform to realize the research motivation. Later, this study investigated the trade-off between system performance and rider comfort of APT passengers in an urban condition, in which the DART system was considered as a case study. The different lateral ride comfort thresholds were used in this study since there was no existing empirical evidence of passenger comfort levels on APT/AB, therefore, the literature data was referenced rationally.

This is the first time that APT platooning and passenger factors were investigated in a microscopic traffic simulation using the human-centric design approach. This method has emerged in recent decades

and is considered as the central concept in developing technology and transportation infrastructure for human beings [49,50]. The study has several limitations which can be improved in future research.

1. Firstly, the simulation scenarios allow to ideally prioritize the merged-platoons from start-to-end to improve travel speed and reduce travel time, but the delay impact on private cars have not been evaluated. This delay can be quite severe, as there can be a long waiting time for the whole APT platoon to pass by, especially during peak hours or in case of longer merged platoons e.g., of 10 modules. The trade-off is now expanding to private car drivers' perceptions and the whole network performance for both APT and private cars, which is more challenging to solve.

2. Secondly, due to a single operational corridor in this study, the delay investigation is not comprehensive. A more extensive network with multiple lines practicing coupling and decoupling, and traffic demand inputs are worth investigating for further study. It is noted that the planned DART network includes 18 lines with vast and complicated coupling/decoupling process across these lines. The performance issues may happen and deteriorate the whole system's reliability when the number of modules within each platoon, the number of APT lines and the network are scaled up.

3. Moreover, the effect of road excitation on passenger comfort, which is also an important influencing factor, has not been considered. For the urban bus, air-suspension is often equipped to maintain the high comfort levels at a lower natural frequency as well as the kneeling function by modifying the internal pressure [51]. It is of utmost crucial importance for passengers on APT/AB (also AV) to enjoy their activities onboard, meaning a smoother road surface is required as compared to the conventional bus system. The bus ride index [52] can be one of the potential solutions to solve this problem.

**Author Contributions:** Conceptualization, T.N.; data curation, T.N. and M.X.; formal analysis, T.N.; funding acquisition, A.R.; investigation, T.N. and M.X.; methodology, T.N.; project administration, T.N., A.R. and W.Y.D.; resources, M.X.; software, N.A.; supervision, B.L., Fritz Busch and W.Y.D.; visualization, T.N. and X.L.; writing–original draft, T.N. and M.X.; writing–review & editing, M.X., A.R., B.L., F.B. and W.Y.D.

**Funding:** This work was supported by the German Research Foundation (DFG) and the Technical University of Munich (TUM) in the framework of the Open Access Publishing Program.

**Acknowledgments:** This work is part of the PhD study of the first author and is financially supported by the Singapore National Research Foundation under its Campus for Research Excellence and Technological Enterprise (CREATE) programme.

**Conflicts of Interest:** The authors declare no conflicts of interest.

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
