# Peer review of "Platooning of Autonomous Public Transport Vehicles: The Influence of Ride Comfort on Travel Delay"

_sustainability, doi:10.3390/su11195237_

Round 1

Reviewer 1 Report

My overall judgement of the paper is very good.
The research work is properly introduced and the pre-existent literature work has been properly referred to and adopted as a source for further development.
Language is clear, the structure and the writing style of the paper are excellent.

I just have 2 comments that maybe could lead to some improvements.

1) Figure 5 could be improved avoiding to be just a screenshot of an excel sheet but being presented as a table, maybe enlarging fonts and just including the data highlighted in red

2) I failed to grasp how the number of modules in the composition of the platoon played a role in the simulations. In other words, how this parameters has been taken into account, mathematically, in the simulation? Since from the starting point to the end the platoon act as a single "train", how the number of the modules can affect its travel time?
I am pretty confident that this aspects is explained in the paper, but I failed to grasp it, therefore it might be that it needs to be better clarified along the text.

Author Response

Response to Reviewer 1 Comments

Dear Reviewer

The authors would like to thank the Reviewer for careful review of our manuscript and providing us with comments and suggestions to improve the quality of the manuscript. Upon carefully considering the Reviewer’s comments, several amendments have been applied correspondingly. The changes are highlighted in the revised manuscript by using red-coloured text. The following responses have been prepared to address all the Reviewer’s comments in a point–by–point fashion

Point 1: Figure 5 could be improved avoiding to be just a screenshot of an excel sheet but being presented as a table, maybe enlarging fonts and just including the data highlighted in red.

Response 1: Figure 5 shows the real-time information of all DART modules existing in the simulated network and this information is updated dynamically by every simulation time step of 1/10 second. We prefer keeping this generated screenshot from the VISSIM simulation, which is a recent development from us to integrate new information of the coupling/decoupling protocol into the current commercial software.

The fonts in Figure 5 are now enlarged in the revised manuscript as per the Reviewer’s suggestion.

Point 2: I failed to grasp how the number of modules in the composition of the platoon played a role in the simulations. In other words, how this parameters has been taken into account, mathematically, in the simulation?

Response 2: The idea to create platoons constituted of a varying number of modules is to test the coupling/decoupling protocol that we developed. The investigation on the influence of ride comfort on travel delay of APT platooning cannot be conducted without functional platooning formation. Herein, we applied different scenarios involving the different number of modules in order to observe the microscopic driving behaviours in platoon formation and observing their trajectories as well as travel time along the corridor.

Each module within the merged-platoon slows down when entering a curve and at some points, the gaps between modules open up, but later on the slower module can speed up and catch up on the one ahead. There is a lot of vehicle-to-vehicle communication within the merged-platoon. Indeed, the formation of platoon length is a challenge in the calculation capability of VISSIM simulation that resulted in the software crashing in scenarios with more than 6 modules in composition.

Point 3: Since from the starting point to the end the platoon act as a single "train", how the number of the modules can affect its travel time? I am pretty confident that this aspects is explained in the paper, but I failed to grasp it, therefore it might be that it needs to be better clarified along the text.

Response 3: It is correct that upon formation, the platoon will travel as a single “train” along the corridor. We hypothesise that there will be dynamical effects of the length of platoon (or the number of modules) on travel time along the corridor. However, it turned out that having 3, 4, or 5 modules in the platoon does not cause any delay effect. In effect, a slower module can speed up to catch the one ahead in order to maintain a certain/desired time gap (t=1.5s in our simulation scenario).

This driving behaviour in a road train differs from normal rail train movement in which many rail coaches are physically connected and the rail train requires much larger turning curve along the dedicated railway. Herein, the DART road train with the electronic connection has greater ability and flexibility to negotiate sharp turning curves.

We highlighted this aspect in the revised manuscript as per the Reviewer’s suggestion. Please refer to the revised part (in lines 236-240) and discussion part (in lines 294-296) in the revised manuscript.

Reviewer 2 Report

The paper presents an interesting research. The paper is well developed and put together. The findings - results and discussion, are also well presented. The list of references is also comprehensive and current. Overall, this paper adds value to the field. 

Author Response

Dear Reviewer

The authors would like to thank the Reviewer for careful review of our manuscript and providing us with comments and suggestions to improve the quality of the manuscript.

Reviewer 3 Report

1-In your model, have you considered how much the AADT is? I was not able to find certain AADT for type of roadways you have considered.

2- Will you be able to use any statistical software packages for modeling your traffic data?

3-on line 289, you have mentioned turning curves. Is there any specific information related to type of curves in your study to include their information?

4- If you could include some results using Arcmap or mentioning in your literature review, the results from the paper entitled "Impact of Advertising Signs on Freeway Crashes within a Certain Distance in Michigan" on how the authors used Arc GIS and defined certain buffer zone in their model, would support your literature review more.

Author Response

Response to Reviewer 3 Comments

Dear Reviewer

The authors would like to thank the Reviewer for careful review of our manuscript and providing us with comments and suggestions to improve the quality of the manuscript. Upon carefully considering the Reviewer’s comments, several amendments have been applied correspondingly. The changes are highlighted in the revised manuscript by using red-coloured text. The following responses have been prepared to address all the Reviewer’s comments in a point–by–point fashion

Point 1: In your model, have you considered how much the AADT is? I was not able to find certain AADT for type of roadways you have considered.

Response 1: In this study, the effect of AADT is not considered since we simulate an ideal scenario of traffic light prioritisation. In the beginning, we included the AADT data and traffic signal data collected from the Land Transport Authority of Singapore, and we did not observe much difference in the results from our simulations with varying AADT (under traffic light prioritisation).

Point 2: Will you be able to use any statistical software packages for modeling your traffic data?

Response 2: The data is processed using ‘R’ software package. This information is now included in the revised manuscript (see lines 234-235).

Point 3: on line 289, you have mentioned turning curves. Is there any specific information related to type of curves in your study to include their information?

Response 3: The detailed turning curves are shown in Figure 3 and Table 4, including six curves in which three radii are less than 100m, and the other three radii are greater than 100m. The reduced speeds were back-calculated following different curve radii (see Equation 1).

We assumed that these are all normal horizontal curves in the urban context (without consideration of super-elevation or transitioning such as spiral or Clothoid curve).

Point 4: If you could include some results using Arcmap or mentioning in your literature review, the results from the paper entitled "Impact of Advertising Signs on Freeway Crashes within a Certain Distance in Michigan" on how the authors used Arc GIS and defined certain buffer zone in their model, would support your literature review more

Response 4: We thank Reviewer for the suggested reference, which is now included in the revised manuscript as regarding the application of AutoCAD for analysing curve radius (lines 216-217).